# Multifaceted Functional Role of Semaphorins in Glioblastoma

**DOI:** 10.3390/ijms20092144

**Published:** 2019-04-30

**Authors:** Cristiana Angelucci, Gina Lama, Gigliola Sica

**Affiliations:** Istituto di Istologia ed Embriologia, Università Cattolica del Sacro Cuore, Fondazione Policlinico Universitario A. Gemelli, IRCCS, 00168 Rome, Italy; gina.lama@unicatt.it (G.L.); gigliola.sica@unicatt.it (G.S.)

**Keywords:** glioblastoma, semaphorins, plexins, neuropilins, cell proliferation, cell migration, cell invasiveness, angiogenesis, immune response

## Abstract

Glioblastoma (GBM) is the most malignant tumor type affecting the adult central nervous system. Despite advances in therapy, the prognosis for patients with GBM remains poor, with a median survival of about 15 months. To date, few treatment options are available and recent trials based on the molecular targeting of some of the GBM hallmark pathways (e.g., angiogenesis) have not produced any significant improvement in overall survival. The urgent need to develop more efficacious targeted therapies has led to a better molecular characterization of GBM, revealing an emerging role of semaphorins in GBM progression. Semphorins are a wide group of membrane-bound and secreted proteins, originally identified as axon guidance cues, signaling through their receptors, neuropilins, and plexins. A number of semaphorin signals involved in the control of axonal growth and navigation during development have been found to furthermore participate in crosstalk with different dysfunctional GBM pathways, controlling tumor cell proliferation, migration, and invasion, as well as tumor angiogenesis or immune response. In this review, we summarize the regulatory activities mediated by semaphorins and their receptors on the oncogenic pathways implicated in GBM growth and invasive/metastatic progression.

## 1. Introduction

Glioblastoma (GBM) is the most frequent and lethal type of tumor affecting adult brain tissue. It is also the highest-grade glioma type (grade IV, WHO) with a predominant astrocytic differentiation. Distinctive pathological features of GBM include heterogeneity in cell size and shape, nuclear atypia, microvascular hyperplasia, and necrotic foci with surrounding cellular pseudopalisades (Figure 1). Despite multimodal aggressive treatment (e.g., surgical resection followed by radiotherapy, with adjuvant temozolomide-based chemotherapy), GBM prognosis remains poor, with a median patient survival of around 15 months [1]. Relapse invariably occurs due to the high local invasiveness of tumor cells (Figure 1) into the healthy neighboring brain tissue. The failure of standard therapy is ascribed to the typical histopathological features of GBM, especially the florid neovasculature, the highly invasive growth pattern, along with cancer cells’ heterogeneity. In particular, tumor progression has been linked to a subset of cancer cells with stem properties, such as self-renewal and pluripotency, associated with high tumorigenic activity and resistance to therapies (also indicated as cancer stem cells, CSCs) [2].

Although our knowledge about the complex network of GBM oncogenic pathways has led to the testing of small-molecule inhibitors or antibodies directed against receptor tyrosine kinases (RTKs; e.g., Ras/MAP/ERK, VEGFR, EGFR, c-Met) or RTK downstream signaling pathways (e.g., PI3K/AKT/mTOR), these treatments have essentially produced disappointing results [3,4]. This raised the need for developing novel and more effective therapeutic strategies for tumor management. An unavoidable condition to achieve this goal is a deeper understanding of the molecular/genetic features of GBM cancer cells, their microenvironment, and peri-tumoral tissue; which is also warranted for better tumor subtype classification and patient stratification. During the past decade, several research groups have worked at the definition of the GBM molecular signature. These studies revealed oncogene and oncosuppressor gene mutations, epigenetic changes, as well as abnormal activation of growth factor receptors or RTKs, to name just a few [5,6,7,8,9,10,11]. Since recurrence in peritumoral tissue arises within 2 cm from the border of the original lesion in about 95% of patients, the molecular features of this area are of particular interest. For instance, our previous studies demonstrated that GBM peritumoral tissue, even when lacking cells with overt neoplastic morphology, expresses the CSC markers CD133 and nestin, as well as typical markers of tumor growth, invasion, and angiogenesis [5,6,7,8]. A rich network of microvessels with aberrant structure and the expression of neoangiogenesis markers CD105, VEGF, and VEGF-Rs have also been observed in the peritumoral GBM tissue. In addition, the analysis of expression signatures revealed an up-regulation of genes associated with cell proliferation and motility/invasiveness not only within GBM, but also in the peritumoral tissue [5,6,7,8]. Altogether, these findings suggest that cancer-promoting signals in GBM-surrounding tissue can foster tumor recurrence. This is consistent with the idea of a potential role of neural guidance molecules in controlling GBM progression. Indeed, the motile and invasive behavior of glioma cells infiltrating the brain parenchyma suggests a regulatory role of guidance cues and/or chemotropic factors, possibly analogous to those active during nerve tissue development [12]. For example, a role for axon guidance molecules such as netrin-1 or Slit-2 has been demonstrated in glioma cell migration [13,14]. Furthermore, semaphorins are a wide family of transmembrane, secreted, or membrane glycosylphosphatidylinositol-linked glycoproteins grouped into eight classes, which were originally identified for their ability to guide axonal growth and navigation in the wiring of developing neuronal circuits [15]. Class 1 and 2 semaphorins include invertebrate semaphorins, classes 3–7 are found in vertebrates, while class-5 semaphorins are encoded by viral genomes. Distinct structural motifs characterize the different semaphorin classes; thus, immunoglobulin-like domains are present in semaphorins of classes 4 and 7, whereas class-5 semaphorins contain thrombospondin repeats. Moreover, class-3 semaphorins, which comprise secreted vertebrate semaphorins, are characterized by a conserved basic-charged motif at the C terminus. However, the other vertebrate transmembrane or membrane-anchored semaphorins can undergo proteolytic cleavage, shedding soluble forms.

Most semaphorin signals are mediated through the interaction with high-affinity receptors of the plexin family [16], which can bind semaphorins either alone or in combination with the co-receptors neuropilins (NRPs). Beyond axon guidance, semaphorin signals affect a variety of other functions, ranging from cell migration and substrate adhesion, to cell viability and proliferation, to immune response and angiogenesis, in both physiological and pathological contexts. The first evidence of the involvement of semaphorins in tumorigenesis was provided by the isolation of semaphorin 3A (*Sema3A*) and *Sema3F* genes in the 3p21 chromosomal region, where a deletion was found in the majority of small-cell lung cancers [17,18]. An increasing amount of experimental evidence collected in the last decade has indicated a relevant role for semaphorins in many types of cancer, including GBM [19,20,21,22,23,24]. The involvement of semaphorins and their receptors in GBM was initially suggested based on their expression in human glioma cells [25]. In this review article, we will focus on the multifaceted pro- or anti-tumorigenic activities of semaphorins found to control GBM progression.

## 2. Role of Semaphorins in GBM Cell Growth and Survival

Among the various hallmarks characterizing aggressive tumors, a major factor influencing prognosis is the indefinite proliferative ability of cancer cells and their resistance to apoptosis.

Growing evidence supports the involvement of certain semaphorins as key regulators of GBM cell growth and survival, as shown by in vitro and in vivo experiments (see Table 1 and Figure 2). Notably, it was reported that the overexpression of secreted Sema3A and Sema3F in human GBM cells U87MG dramatically reduced their proliferation, as well as colony formation in soft agar, while no significant variations were observed in culture upon expression of the homologous family members Sema3B, Sema3D, and Sema3G [26]. A tumor-growth-inhibiting activity of Sema3A has also been demonstrated in vivo in this experimental setting, since its expression in U87MG cells implanted in the mouse brain cortex strikingly repressed tumor development; similar tumor-suppressive effects were observed when GBM cells engineered to express Sema3B, Sema3D, Sema3E, or Sema3F were transplanted either in the brain or subcutaneously [26], which is also potentially consistent with their activity in the tumor microenvironment. However, based on other data, the functional role of Sema3A in GBM appears controversial. For instance, while lacking any proliferative or cytotoxic activity in rat C6 GBM cells [27], endogenous Sema3A was found instead to sustain the growth of GBM patient-derived cells (PDCs) [28]. Indeed, a dramatic reduction of cell proliferation was observed in this experimental setting for PDCs treated with the anti-Sema3A antibody F11, compared to IgG-treated controls [28]. The same treatment led to a significant inhibition of tumor growth an in vivo study of mouse xenograft models established via subcutaneous injection of these GBM PDCs [28,29]. A concomitant depletion of Sema3A, decreased phospho-ERK levels, as well as a sharp induction of apoptotic cell death were observed in F11-treated tumors compared to controls [28]. The apparent discrepancy between these data on Sema3A activity in GBM may be consistent with its putative ability to engage different receptor complexes and regulate multiple cell types in the tumor microenvironment, also depending on expression levels. For instance, Bagci et al. proposed that Sema3A could function according to a dose-dependent “biphasic model” [30]. It is thus conceivable that recombinant Sema3A overexpression in GBM cells could lead to different signaling mechanisms and functional effects compared to those deployed at endogenous levels.

Other studies conducted in GBM U87MG and U251 cells demonstrated that the blockade of Sema3B-targeted miRNA miR-221 led to an upregulation of Sema3B protein levels, which was associated with decreased cancer cell proliferation [31]. An anti-tumorigenic activity of Sema3F was also observed when GBM cells were inoculated subcutaneously in mice treated systemically with a single intravenous injection of adenoviral particles encoding this semaphorin [32]. This tumor-suppressing effect induced by either local or systemic administration of Sema3F was attributed to Sema3F/NRP2-mediated inhibition of Akt-mTOR signaling [32]. Although promising, these in vivo findings leave open the question of whether local or systemic administration of the soluble forms of Sema3A, Sema3D, Sema3E, or Sema3F may recapitulate the tumor-suppressing effect of ectopically expressed molecules. Moreover, despite the high leakiness of the tumor blood vessel basement membrane, the presence of the blood–brain barrier may hinder or reduce the distribution of circulating Semas into the GBM microenvironment.

A prosurvival activity of Sema3C was reported in GBM stem cells (GSCs) isolated from human specimens. In particular, endogenous Sema3C promoted GSCs survival and self-renewal via interaction with Plexin-A2/D1 receptor complexes that, in turn, activated Rac1 pathway [33]. This signaling cascade seems to be preferentially used by GSCs, while in non-stem tumor cells derived from the same GBM patients no expression of Sema3C or Plexin-A2/D1 receptors has been detected [33].

As concerning class-4 semaphorins, indirect experimental evidence suggested the involvement of Sema4D in GBM progression, as a stable silencing of its receptor, Plexin-B1, in human U87MG and U251 GBM cells significantly increased apoptotic death. This effect was mediated by the ability of Plexin-B1 to activate serine-arginine protein kinase 1 (SRPK1) that, when repressed, was found to promote apoptosis [34]. Moreover, Sema4D-activated Plexin-B1 has been shown to elicit the tyrosine kinase activity of Met, the receptor for the hepatocyte growth factor/scatter factor (HGF/SF), whose signals mediate the invasive growth of epithelial cells [35]. A similar mechanism has also been suggested to support GBM progression. In fact, high NRP1 levels in U87MG cells have been found to promote survival and proliferation through the enhancement of autocrine HGF/SF-Met signaling [36].

Among class-6 semaphorins, a role in GBM cell proliferation and tumorigenesis has been demonstrated for Sema6B and its receptor Plexin-A4. In fact, Sema6B or Plexin-A4 knockdown in U87MG cells significantly impaired proliferation in vitro and tumorigenesis upon subcutaneous injection in nude mice [37].

## 3. Role of Semaphorin Signaling in GBM Cell Migration and Invasiveness

GBM is the most invasive primary brain tumor. The marked ability of tumor cells to infiltrate and rapidly invade surrounding tissues is crucial to account for the high frequency of relapse which characterizes this disease. Diverse semaphorins seem to play a relevant regulatory function in the sequence of events leading to GBM cell migration and invasion, displaying either inhibitory or promoting effects (see Table 1 and Figure 2). Sema3A, for example, which is generally known as an anti-tumorigenic molecule, seems to have a rather ambivalent role in GBM, and displays a receptor-dependent functional plasticity leading to opposite migratory behavior of tumor cells. For instance, Sema3A was reported to inhibit GBM cell migration through ABL2 kinase-dependent inhibition of RhoA GTPase [38]. Moreover, recombinant Sema3A was found to repel rat C6 GBM cell migration; however, upon blockade of its receptor complex formed by NRP1/Plexin-A1 this activity was unexpectedly switched to NRP2-dependent chemoattraction [27]. Notably, several studies have indicated that endogenous Sema3A can mediate autocrine signaling in GBM PDCs promoting (rather than inhibiting) cell motility; in fact, in both PDCs and immortalized U87MG cells, Sema3A neutralization with different anti-Sema3A antibodies significantly reduced cell migration [28]. NRP1-dependent autocrine Sema3A signaling was found to promote GBM cell motility through a pathway dependent on Rac1 GTPase activity; in fact, RNA-interference-mediated knockdown of Sema3A in this setting resulted in the inhibition of GBM cell migration and dispersal [30]. Rac1 activation elicits various ultrastructural changes, such as a reduction in stress fibers and the development of membrane ruffles at the leading edge, which enable tumor cell invasion of the extracellular matrix [39]. Interestingly, Bagci and coworkers [30] provided evidence that GBM cells engineered to overexpress Sema3A, displayed reduced scattering instead of the cell spreading promoted by the endogenous molecule; thus, these authors suggested a dose-dependent “biphasic model” for GBM cell control by this signal. In sum, while in other tumor types, endogenous Sema3A levels are often suppressed and its reintroduction deploys inhibitory activity, autocrine Sema3A in GBM cells seems to act instead as promoter of migration via distinctive signaling cascades. In contrast, NRP2/Plexin-A1-mediated Sema3F signals resulted in a rapid collapse of F-actin cytoskeleton and loss of contractility, leading to an impairment of cell motility [39]. These effects have been associated with the activation of the membrane-anchored non-receptor-type tyrosine kinase ABL2 through direct binding to Plexin-A1, which in turn inactivates RhoA—a GTPase responsible for actin polymerization and cell contractility [40]. This key step in the signaling cascade could be blocked by using ABL2 kinase inhibitor imatinib [38]. Further studies confirmed that the inhibitory activity of Sema3F on GBM cell migration depends on NRP2 expression, suppressed by hypoxia in the tumor context, and on the subsequent RhoA inactivation [41].

Sema3B upregulation achieved by the knockdown of regulatory miR-221 empowered the tumor-suppressor activity of this semaphorin in GBM [31,42], leading to the reduced migration and invasiveness of U87MG cells [31]. Similar effects have been demonstrated for Sema3G in U251MG cells [43]; this activity seems to be related to the inhibition of matrix metalloproteinase-2 expression [43]. Among class-4 semaphorins, a possible involvement of Sema4C and Sema4D in controlling GBM cell motility has been documented. Sema4C bound to its receptor Plexin-B2 was found to induce ultrastructural changes in actin cytoskeleton, promoting human U87MG and LN229 GBM cell migration and invasion in vitro and in vivo [44]. RhoA and Rac1 GTPases have been identified as downstream effectors of this signaling cascade, also revealing a synergism between Plexin-B2 and Met tyrosine kinase in the activation of the HGF/SF-Met pathway, as a possible effector of the pro-migratory activity of Sema4C [44]. Moreover, the silencing of the Sema4D receptor, Plexin-B1, in U87MG and U251 cells significantly reduced motility and invasiveness in vitro [34]. Plexin-B1′s pro-migratory/invasive activity is mediated by the regulation of RhoA (found in association with the Plexin-B1 intracellular domain), which activates a downstream signaling pathway leading to cytoskeletal remodeling [34].

Also, the Plexin-B3 Sema5A receptor and the homologous member Plexin-B2 were found to interact with Met tyrosine kinase, mediating the activity of cell-scattering factor HGF [45]. Sema5A/Plexin-B3 interplay induced a significant impairment of cell migration and invasion of rat C6 and human U87MG GBM cells via the direct interaction of Plexin-B3 with RhoGDIα, leading to Rac1 GTPase inactivation. Along with the reduction of the activity of the actin binding protein Fascin-1, Rac1 inactivation resulted in loss of lamellipodia and impairment of cell motility [46,47]. Sema5A expression analysis in human astrocytomas revealed a marked decline of its levels in GBM compared to low-grade tumors, possibly repealing its suppressive activity on cell motility [47]. An inhibitory effect on GBM cell migration and invasiveness was also demonstrated for Sema6A [48]. These migration/invasion inhibitory effects have been attributed to the known inhibitory activity exerted by the Sema6A extracellular domain on ERK1/2 and FAK phosphorylation [48]. Finally, Sema7A expression was found in U87MG GBM cells, but not in other less-invasive glioma cell lines [49]. Previous studies indicated the loss of Sema7A-receptor Plexin-C1 expression to be associated with melanoma progression [50]. Likewise, Plexin-C1 was not detected in U87MG cells where the observed expression of the alternative Sema7A-receptor integrin-β1 suggests that this switch in receptor components may contribute to the tumor cell invasive phenotype [49].

## 4. Role of Semaphorins in GBM Angiogenesis

GBM are highly vascularized tumors; in fact, the presence of a remarkable microvascular proliferation represents a key element and diagnostic hallmark of human GBM, distinguishing this neoplasm from lower-grade gliomas [51,52]. Several mechanisms are involved in the prominent development of blood vessels observed in these tumors and in their peritumoral tissue [53,54]. One is the sprouting of new capillaries from pre-existing blood vessels due to endothelial cell proliferation, which seems to be driven by signals released by the hypoxic tumor core. Indeed, hypoxia-inducible factors, which enhance VEGF expression, are strongly activated in GBM and in peritumoral tissues [55,56].

The involvement of semaphorins, particularly those of class 3, in the regulation of angiogenesis and vascular homeostasis was originally endorsed by the discovery that their co-receptor NRP1 also acts as a co-receptor for VEGF_165_ [57]. It has been shown that NRP1 can modulate VEGF_165_ binding to KDR/Flk1 (VEGF receptor 2, VEGFR2), the major transducer of VEGF signals in endothelial cells, thus controlling VEGF-triggered angiogenesis [58]. Notably, the NRP1 ligand Sema3A displays an inhibitory activity upon developmental angiogenesis, and it also hinders new blood vessel formation in many types of solid tumors [59,60,61,62]. This anti-angiogenic activity may be partly ascribed to the ability of Sema3A to elicit endothelial cell apoptosis after prolonged exposure [63]. Indeed, in GBM, the anti-angiogenic ability of Sema3A matches with its ability to inhibit tumorigenesis in certain experimental settings. In fact, Sema3A-overexpressing U87MG cells were poorly tumorigenic in mice, which was associated with a lower abundance of cancer blood vessels [26]. Interestingly, the overexpression of other class-3 semaphorins which are known NRP1 ligands (i.e., Sema3D and Sema3E) led to the same effect. The same was true of Sema3F, which interacts with NRP2 and inhibits tumor formation and angiogenesis [26]. In association with an inhibition of tumor growth, Sema3F overexpression in subcutaneously injected U87MG cells led to a striking perturbation of angiogenesis, characterized by constricted vessels with collapsed lumens [32]. A single intravenous injection of adenovirus encoding human Sema3F produced comparable effects on the vasculature of pre-implanted U87MG tumor cells [32]. In vitro experimental evidence indicated that the inhibition of aberrant angiogenesis elicited by local or systemic administration of Sema3F in vivo may be mediated by the NRP2-dependent inhibition of VEGF production and Akt-mTOR signaling in tumor cells [32]. Moreover, the hypoxia-induced repression of NRP2 expression in GBM U87MG cells has been found to enhance VEGF paracrine secretion and counteract Sema3F-dependent angiogenesis inhibition [41].

Among the secreted semaphorins, Sema3G did not show any anti-tumorigenic activity and failed to reduce blood vessel density [26].

All class-3 semaphorins can be cleaved by furin-like pro-protein convertases, which are overexpressed in tumor cells [64], resulting in an attenuation of their anti-angiogenic effects, as clearly demonstrated for Sema3B in breast and lung cancer models [65]. Since the class-3 semaphorin-mediated inhibitory effect on developmental and tumor angiogenesis has been associated with their ability to counteract VEGF_165_ signaling, a prognostic value of determining VEGF/Sema3 signal ratio has been suggested in GBM [66].

Notably, beyond its activity on blood vessel formation, Sema3A delivered by extracellular vesicles of patient-derived GBM cells has been found to enhance vascular permeability by disrupting the endothelial barrier integrity, thus contributing to the formation of the typical peritumoral tissue edema [67]. Moreover, Sema3C/Sema3F signaling via NRP2 may regulate lymphoangiogenesis in GBM, as anti-NRP2 treatment significantly reduced lymphatic vessel density in GBM xenografts generated by the injection of rat C6 cells [68].

Sema4D is a membrane bound class-4 semaphorin that binds to the Plexin-B1 receptor [16]. Stable silencing of Plexin-B1 impaired Sema4D-mediated activation of the RhoA and SRPK1 signaling in tumor cells, and resulted in a reduction of microvessel density in xenografts generated in vivo, due to an unknown mechanism [34]. Interestingly, Plexin-B2 knockdown in intracranial U87MG transplants also resulted in markedly reduced tumor vascularization [44]. According to a proposed model for the development of GBM vascularization, tumor cells migrate into the peritumoral area to surround pre-existing brain microvessels, which become progressively enclosed within the growing tumor mass. This mechanism leads to the development of well-vascularized tumors in a neoangiogenesis-independent manner [69]. According to this model, the inhibition of angiogenesis observed in Plexin-B2-depleted GBM xenografts in mice may be due to reduced tumor cell perivascular spread associated with the disruption of the Sema4C/Plexin-B2-mediated pro-migratory pathway [44]. Consistent with this mechanism, it was suggested that the interaction between Plexin-B2 expressed by GBM cells and Sema4C present on neighboring endothelial cells may account for enhanced tumor angiogenesis [44].

Concerning the class-6 semaphorins, silencing the expression of either Sema6B or its receptor Plexin-A4 [16] in U87MG cells has been found to induce a striking inhibition of the tumor-forming ability [37]. Despite the significant inhibitory effect that Sema6B or Plexin-A4 knockdown produced on HUVEC proliferation and tubulogenesis in in vitro experiments, the microvessel density as well as the pericyte coverage of the blood vessel observed in the small tumors arising from Plexin-A4- or Sema6B-silenced U87MG cells were similar to those found in tumors formed by control cells [37]. These findings suggest that the tumorigenic activity of Sema6B may be independent of its ability to affect angiogenesis.

## 5. Role of Semaphorins in GBM Progression by Modulation of Immune Response

GBMs are among the most treatment-resistant human tumors. Over the past two decades, a variety of therapeutic strategies have been developed in an attempt to overcome the limitations and the overall inefficacy of standard treatments, although with poor success. Among the most promising and challenging new therapeutic approaches are those based on the modulation of the host immune system [70,71], as convincing evidence proved the key role played by innate and adaptive immunity in cancer progression [72]. In fact, a massive infiltration of myeloid-derived cells characterizes GBM [73]. Among these, the so called glioma-associated microglia and macrophages (GAMMs) are the dominant tumor-infiltrating immune cells in the GBM microenvironment, playing a central role in the regulation of the antitumor immune response [74]. Tumor microglia originate from resident CNS macrophages, while circulating monocytes gives rise to tumor-associated macrophages (TAMs). GAMMs are recruited to the GBM site, where they do not carry out any immune effector function, but instead promote tumor growth and invasiveness [72]. This switch in GAMM function may be partly mediated by the release of macrophage inhibitory cytokine-1 (MIC-1) by GBM stem cells [74]; in turn, this factor induces GAMMs to secrete immunosuppressive cytokines such as transforming growth factor-β1 [75].

The expression of several semaphorins and their cognate receptors has been described in a variety of lymphoid and myeloid immune cells, controlling their growth, differentiation, chemotaxis, and cytokine release [76,77]. Experimental evidence proved sempahorins’ role in the modulation of the immune response in various cancers, with particular reference to Sema3A and Sema4D. Sema3A seems to interfere with the detection of neoplastic cells by the immune system. It induces the apoptosis of monocyte-derived macrophages, while tumor-cell-secreted Sema3A has been found to inhibit primary human T-cell proliferation and cytokine secretion [78,79]. Indeed, Sema3A-silenced lung and renal tumor cells lose this T-cell blocking activity, which is mediated by the repression of Ras/mitogen activated protein kinase (MAPK) signaling [79]. Accordingly, Plexin-A4—a component of the receptor complex for class-3 semaphorins which is also expressed in the lymphoid tissues—has been shown to mediate a negative regulation of T-cell-mediated immune responses [80]. The role of NRP1 expressed by TAMs in controlling their entry into tumor hypoxic regions in response to Sema3A has been investigated [81]. Notably, *NRP1* gene deletion in TAMs resulted in their entrapment in normoxic tumor regions and lack of infiltration in the tumor core, with a consequent reduction of their pro-angiogenic and immunosuppressive activities. These data indicate that Sema3A/NRP1 signaling may guide the migration of TAMs in hypoxic niches to escape antitumor immune surveillance and promote tumor progression [81].

Consistent with these findings, it was recently shown that the recruitment of TAMs was significantly reduced compared to controls in GBM-patient-derived xenograft models treated with anti-Sema3A antibody [28]. These findings suggest that the Sema3A/NRP1 signaling blockade in GBM might prevent TAM infiltration and underscore the use of anti-Sema3A antibodies as a potential therapeutic approach [28]. Beyond Sema3A/NRP1 signals, the Sema4D/Plexin-B1 interaction also seems to play a significant role in modulating the immune response in GBM. Indeed, Sema4D (also indicated as CD100) was detected in natural killer (NK) cells. Co-cultures of NK and GBM cells at different effector/target ratios revealed a progressive increase in NK cytotoxicity [82]. Thus, interaction between Sema4D expressed in GBM cells and Plexin-B1-expressing NK cells seems to promote the NK-cell-mediated killing of tumor cells [82]. As discussed above, GAMMs’ pro-tumorigenic activity represents a major element in the microenvironment of all GBM subtypes, and their proved role in tumor progression has made them an attractive target for pre-clinical studies [83,84,85,86]. Interestingly, the monocyte/macrophage population in human GBM samples significantly correlates with NRP1 expression [87]. Mice lacking NRP1, which is known to control GAMM polarization, display reduced GBM volume and vascularization, with a parallel increase in anti-tumorigenic GAMM infiltrate [88]. The replacement of peripheral macrophages with NRP1-silenced macrophages derived from bone marrow inhibits GBM development [88]. In mice with NRP1-depleted microglia and wild-type peripheral macrophages, repressed tumorigenesis was also observed with increased microglial infiltration, compared to mice with wild-type GAMMs [88]. These results globally indicate that NRP1 removal in GBM GAMMs results in a reprogramming of their phenotype, also impairing their proangiogenic and immunosuppressive activity [88].

Recent experimental evidence obtained in animal models [89] has led to a first phase I clinical trial with a humanized anti-Sema4D antibody in patients with advanced solid tumors, demonstrating its good tolerability; moreover, 45% of patients exhibited a stable disease of ≥8 weeks [90]. However, the implicated effector mechanisms are not completely elucidated.

## 6. Concluding Remarks

Despite great advances in our knowledge of cancer biology, GBM remains a dreadful disease with very poor prognosis. Resistance to conventional therapies can be considered a hallmark of GBM, due to its complex and heterogeneous molecular signature [91]. All newly diagnosed GBMs are treated with the same standard-of-care, consisting of maximal surgical resection, followed by radiotherapy and temozolomide-based chemotherapy. However, GBM’s great variability at the molecular and clinical levels has raised the need for tailored therapeutic approaches that can meet the specific features of GBM subtypes. Over the past decade, an increasingly detailed definition of the complex network of interplaying oncogenic pathways contributing to abnormal tumor cell functions has led to the development of different targeted therapies, such as those suppressing kinase activities (e.g., c-Met, PI3K, VEGFR), which unfortunately have generally yielded disappointing results in GBM. In this scenario, increasing evidence indicates that semaphorins and their receptors are deeply involved in the regulation of GBM progression [92]. Semaphorins provide pro-tumorigenic stimuli to GBM cells, contributing to their aberrant growth, increased migratory ability and invasiveness, and/or to enhanced angiogenesis, and may represent promising therapeutic targets. Conversely, other semaphorin signals may be exploited as GBM-suppressor molecules. Notably, in a first-in-human phase I study of patients with advanced solid tumors, the treatment with the VX15/2503 anti-Sema4D antibody was well tolerated and produced the expected pharmacodynamic effects [90]. At present, various phase I or phase I/II clinical trials with VX15/2503 in combination with other immune checkpoint inhibitors are in progress to evaluate safety, tolerability, efficacy, and biological endpoints in patients affected by tumors other than GBM (i.e., colorectal, pancreatic, mammary, melanoma, osteosarcoma). Additional studies are required to elucidate the relevance of apparently contradictory effects produced by certain semaphorins, such as Sema3A, in different GBM experimental settings, and better delineate their actual implications in the future of GBM therapy.

## Figures and Tables

**Figure 1 ijms-20-02144-f001:**
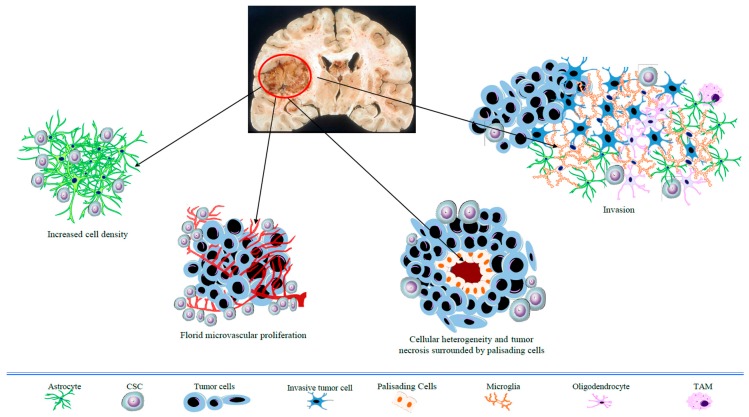
Schematic overview of the histopathological hallmarks of glioblastoma (GBM). CSC: cancer stem cell; TAM: tumor-associated macrophage.

**Figure 2 ijms-20-02144-f002:**
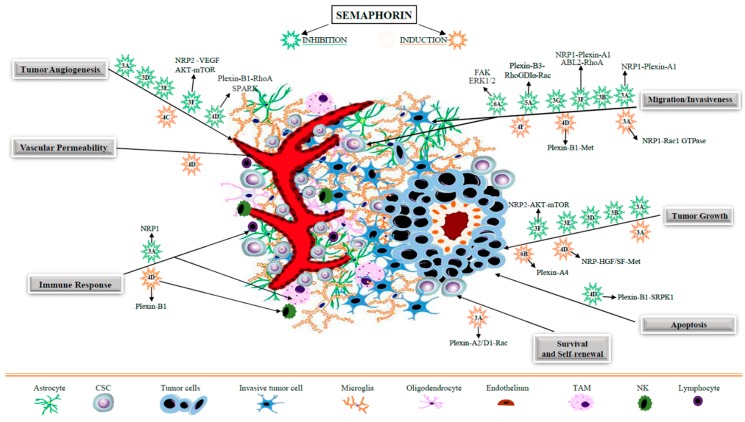
Semaphorin signals regulating GBM cells and the tumor microenvironment. The figure summarizes pro- or anti-tumorigenic activities mediated by diverse semaphorins controlling GBM progression by the regulation of diverse cell populations in the tumor microenvironment. Divergent semaphorin functions may be explained by specific plexin–neuropilin (NRP) receptor complexes or intracellular effector pathways (also indicated, as reported in literature).

**Table 1 ijms-20-02144-t001:** Current knowledge about the function of diverse semaphorins in GBM.

Semaphorin	Pro-Tumorigenic	Anti-Tumorigenic
Sema3A	Increased growth of GBM patient-derived cells (PDCs), in vitro and in vivo [28,29]. NRP1 mediated induction of human A172 and U87MG cell migration via Rac1 activation [30]. Increased migration of PDCs and U87MG cells [27]. Induced recruitment of TAMs [28].	Reduced proliferation and colony formation of U87MG cells in vitro and inhibition of tumor development in vivo [26]. Inhibition of U87MG cell motility mediated by RhoA inactivation [38]. Inhibition of angiogenesis in U87MG tumors in vivo [26].
Sema3B	_	Inhibition of U87MG tumor development in vivo [26]. Reduction of U87MG and U251MG cell proliferation, migration and invasiveness [31].
Sema3C	Increased survival and self-renewal of GBM stem cells (GSCs) obtained from tumor specimens via Plexin-A2/D1-mediated Rac1 signaling activation [33].	_
Sema3D	_	Inhibition of U87MG tumor development and angiogenesis in vivo [26].
Sema3E	_	Inhibition of U87MG tumor development and angiogenesis in vivo [26].
Sema3F	_	Reduced proliferation and colony formation of U87MG cells in vitro [26], and inhibition of tumor development and angiogenesis in vivo [26,32]. NRP2/Plexin-A1 signaling-mediated inhibition of U87MG cell motility via RhoA inactivation [38,39,40,41]. Hypoxia-induced repression of NRP2 expression in GBM U87MG enhanced Sema3F-mediated endothelial cell migration and sprouting [41].
Sema3G		Reduction of U251MG cell migration and invasiveness mediated by the inhibition of metalloproteinase-2 expression [43].
Sema4C	Increased U87MG and LN229 cell migration in vitro through RhoA and Rac1 activation [44]. Increased tumor growth, invasion, and angiogenesis, and tumor cell perivascular spread via Plexin-B2 signaling in vivo [44].	_
Sema4D	Increased U87MG and U251 cell viability, migration, and invasiveness via Plexin-B1 signaling [34]. Increased angiogenesis in U87MG xenograft tumors [34].	Plexin-B1 mediated promotion of natural killer (NK)-cell-mediated killing of tumor cells [82].
Sema5A	_	Reduction of rat C6 and human U87MG cell migration and invasion via Plexin-B3 mediated Rac1 inactivation [46,47].
Sema6A	_	Inhibition of U87MG and U251 cell migration and invasion in vitro, probably mediated by Sema6A-induced suppression of ERK1/2 and FAK activation [48].
Sema6B	Plexin-A4-mediated stimulation of U87MG cell proliferation in vitro and tumorigenesis in nude mice [37].	_

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
