# Peer review of "Multifaceted Functional Role of Semaphorins in Glioblastoma"

_ijms, 2019, doi:10.3390/ijms20092144_

Reviewer 1 Report

The manuscript is well structured, but is not edited very well. There are various formatting styles, even in the same phrase that makes the article hard to read. The table 1 looks like a printscreen and it has an incorrect style. It needs to be corrected. The article needs some more tables and some schemes, figures to be easier to understand and follow.

The authors should add more on their input (their own research on the field, their opinion on the future directions). I noticed just this statement “For about two decades, several research groups have worked at the definition of a GBM molecular signature [5-11]” I don’t think that this adds anything new and is just a way to add self-citations.

The article should include a section on the drugs designed to target this signaling path (small molecules, antibodies etc). This section could be developed to offer a better prediction for the future of the glioblastoma treatment.

Is not Gleevec, this is a proprietary name, it should be imatinib. Please correct all other cases.

I recommend the author to re-think the manuscript not only for the people with a very good background in cell signaling, but also for scientists with various other backgrounds. As said before, some more tables and some figures seem essential for a better understanding of the described problems.

Author Response

Dear Editor,

Please find enclosed our revised manuscript entitled “Multifaceted Functional Role of Semaphorins in Glioblastoma[ijms-478374] by C. Angelucci, G. Lama, and G. Sica, for publication consideration in the Special Issue "Semaphorins, Plexins, and Neuropilin Functions in the Tumor Microenvironment" of the International Journal of Molecular Sciences.

We are grateful to the editors and peer reviewers for their thoughtful comments that helped us to improve the quality of our manuscript.

We answered all the reviewers’ comments and modified the manuscript accordingly. We hope you will find our changes satisfactory. Please find below our responses (normal font) to the reviewers’ comments (italic font).

Reviewer #1:

The manuscript is well structured, but is not edited very well. There are various formatting styles, even in the same phrase that makes the article hard to read. The table 1 looks like a print screen and it has an incorrect style. It needs to be corrected. The article needs some more tables and some schemes, figures to be easier to understand and follow.

We have carefully corrected the font and style throughout the manuscript (including Table 1). As requested by the reviewer, two new Figures have been included.
In particular, we hope that Figure 2, together with Table 1, will help to understand and follow more easily the content of the text.

The authors should add more on their input (their own research on the field, their opinion on the future directions). I noticed just this statement “For about two decades, several research groups have worked at the definition of a GBM molecular signature [5-11]” I don’t think that this adds anything new and is just a way to add self-citations.”

As suggested by the Reviewer, we have extensively modified the Introduction, adding more information regarding our findings and research contribution in GBM and peritumoral tissue characterization.

“The article should include a section on the drugs designed to target this signaling path (small molecules, antibodies etc). This section could be developed to offer a better prediction for the future of the glioblastoma treatment.”

To our knowledge, due to the novelty of this field, very few potential drugs targeting semaphorin signalling have been assessed up to date in preclinical studies, and none of them in the context of GBM models. Notably, clinical studies (phase 1 or phase 1/2 trials) are underway for the evaluation of VX15/2503 anti-Sema4D antibody as single agent or in combination with other immunotherapies for the treatment of diverse tumors other than GBM (i.e. colorectal, pancreatic, and mammary carcinoma, melanoma, osteosarcoma) (Evans EE et al, Proceedings of AACR Annual Meeting 2018). In the first-in-human study of patients with advanced solid tumors, VX15/2503 treatment was well tolerated and produced expected pharmacodynamic effects (Patnaik A et al, Clin Cancer Res, 2016). These data have now been mentioned in the section of concluding remarks, as proof of evidence of semaphorin targeting in translational medicine perspective. Nevertheless, considering the paucity of experimental evidence to be discussed, we feel that the inclusion in our manuscript of a section focusing on this particular aspect would not significantly improve our survey on the role of semaphorins in GBM.

Is not Gleevec, this is a proprietary name, it should be imatinib. Please correct all other cases.”

Thank you for pointing this out. “Imatinib” instead of “Gleevec” is now mentioned in the text.

“I recommend the author to re-think the manuscript not only for the people with a very good background in cell signaling, but also for scientists with various other backgrounds. As said before, some more tables and some figures seem essential for a better understanding of the described problems.”

We understand the reviewer’s concern, and tried to improve the clarity of our message for a general readership by enriching the manuscript introduction and references, as well as including two new explanatory figures. Notably, since this manuscript was intended for consideration within a Special Issue of International Journal of Molecular Sciences, specifically dedicated to the function of semaphorins and their receptors in the tumor microenvironment (Editor: Danna Ann), we are confident that the keynote included in the Special Isssue and other more general articles will provide a detailed background to potential readers coming from different research fields.

On behalf of all the co-authors, I wish the revised manuscript is now sufficiently improved to deserve publication in the Special Issue of IJMS on "Semaphorins, Plexins, and Neuropilin Functions in the Tumor Microenvironment".

Yours sincerely,

Cristiana Angelucci

Reviewer 2 Report

The manuscript by Angelucci et al. presents current knowledge on the role of semaphorins in glioblastoma. The paper is well written and logically structured. However, most of the studies cited in this manuscript are descriptive and provide relatively little mechanistic insights. Recently, an exhaustive review presenting the role of plexins and semaphorins in glioma has been published (Emerging role of plexins signaling in glioma progression and therapy. Angelopoulou E, Piperi C. Cancer Lett. 2018 Feb 1;414:81-87. doi: 10.1016/j.canlet.2017.11.010. Epub 2017 Nov 10), therefore it is difficult to find novelty in this manuscript.     

Major comments:

1). The last paragraph of the Introduction suggests that the review is focused on the pathways deregulated in GBM that involve semaphorins and their receptors. However, this review presents data on the putative roles of semaphorins in GBM and the pathways that can be involved in the function of semaphorins.

2). The weak point of this manuscript is the paucity of definitive data from the literature.  Examples cited in this review are: “In disagreement with these results…“ (p. 2, l. 44); “…Sem3A, generally characterized as an anti-tumorigenic molecule, but whose role in GBM seems rather ambivalent and controversial” (p. 4, l.  28); “…which is in contradiction with other studies described above” (p. 5, l. 3). A greater degree of interpretation of the causes of these discrepancies would improve the manuscript.

3). Do the data presented support a causative or correlative role of semaphorins in GBM?

4). It would be reasonable to present a figure/s related to the mechanism of action of semaphorins.

Minor comments:

Page 4, l. 25-27:  somehow this sentence is unclear, please rewrite it.

Page 5, l. 10: it would be reasonable to add “to” after “related”

Page 5, l. 25: “serve” would be more reasonable (instead of “serves”)

The letters in the Tab. 1 are blurred.

It seems that different fonts were used throughout the manuscript.

Author Response

Dear Editor,

Please find enclosed our revised manuscript entitled “Multifaceted Functional Role of Semaphorins in Glioblastoma[ijms-478374] by C. Angelucci, G. Lama, and G. Sica, for publication consideration in the Special Issue "Semaphorins, Plexins, and Neuropilin Functions in the Tumor Microenvironment" of the International Journal of Molecular Sciences.

We are grateful to the editors and peer reviewers for their thoughtful comments that helped us to improve the quality of our manuscript.

We answered all the reviewers’ comments and modified the manuscript accordingly. We hope you will find our changes satisfactory. Please find below our responses (normal font) to the reviewers’ comments (italic font).

Reviewer #2:

The manuscript by Angelucci et al. presents current knowledge on the role of semaphorins in glioblastoma. The paper is well written and logically structured. However, most of the studies cited in this manuscript are descriptive and provide relatively little mechanistic insights. Recently, an exhaustive review presenting the role of plexins and semaphorins in glioma has been published (Emerging role of plexins signaling in glioma progression and therapy. Angelopoulou E, Piperi C. Cancer Lett. 2018 Feb 1;414:81-87. doi: 10.1016/j.canlet.2017.11.010. Epub 2017 Nov 10), therefore it is difficult to find novelty in this manuscript.”

We respect the opinion of the Reviewer and yet hope that our revised manuscript can contribute significantly to knowledge dissemination by providing an updated survey of the scientific literature, after the article published by Angelopoulou et al., which focused on the signalling mechanisms mediated in GBM by plexins, the main semaphorin receptors. In our revised paper, we decided to put a focus also on the role of neuropilins, which act as co-receptors for class-3 semaphorins, as well as for other growth factors, such as VEGF, TGF-β or HGF. Thus, neuropilins, in addition to associating with plexins, also form complexes with other membrane-anchored receptors, leading to signalling interplays discussed in our manuscript, especially regarding semaphorin function in tumor cell migration and angiogenesis. Moreover, the review article by Angelopoulou et al., despite the wealth of relevant detailed information, for chronological reasons (it was accepted for publication in 2017) could not report the data of many recent studies (8 published in 2018) which are mentioned in our manuscript.

Major comments:

“The last paragraph of the Introduction suggests that the review is focused on the pathways deregulated in GBM that involve semaphorins and their receptors. However, this review presents data on the putative roles of semaphorins in GBM and the pathways that can be involved in the function of semaphorins.”

According to rightful Reviewer’s comment, the last paragraph of the Introduction has been modified as follow: “In this review article, we will focus on the multifaceted pro- or anti-tumorigenic activities of semaphorins found to control GBM progression.”

 “The weak point of this manuscript is the paucity of definitive data from the literature.  Examples cited in this review are: “In disagreement with these results…“ (p. 2, l. 44); “…Sem3A, generally characterized as an anti-tumorigenic molecule, but whose role in GBM seems rather ambivalent and controversial” (p. 4, l.  28); “…which is in contradiction with other studies described above” (p. 5, l. 3). A greater degree of interpretation of the causes of these discrepancies would improve the manuscript.”

We appreciate the insightful comment of the reviewer and did our best to address it in the revised manuscript. The presence of controversial data in the literature is a fact; however, whenever reasonable, a putative interpretation of the discrepancies has been provided. In particular, we discussed the potential reasons accounting for divergent effects mediated by Sema3A in GBM cells reported by different authors (especially concerning proliferative and migratory responses).

“Do the data presented support a causative or correlative role of semaphorins in GBM?”

In our review article we have discussed numerous studies based on mechanistic experiments in cultured cells or in vivo mouse models, which indicate a causative role of semaphorin signals in GBM.

“It would be reasonable to present a figure/s related to the mechanism of action of semaphorins.”

A new figure (Figure 2), summarizing the activity of semaphorins in the different cell types found in the GBM microenvironment, has now been included.

Minor comments:

“Page 4, l. 25-27: somehow this sentence is unclear, please rewrite it.”

As requested by the Referee, the sentence has been rewritten in a more comprehensible form.

“Page 5, l. 10: it would be reasonable to add “to” after “related””

The requested change has been made.

“Page 5, l. 25: “serve” would be more reasonable (instead of “serves”)”

The verb “serves” was referring to the singular term “decline”. However, we decided to rephrase this statement for clarity.

“The letters in the Tab. 1 are blurred.”

Table 1 was reloaded in the file, so that it should be readable now.

“It seems that different fonts were used throughout the manuscript.”

We cannot understand the cause of this defect. We carefully checked now that a single font (Times New Romans) is used throughout the manuscript.

On behalf of all the co-authors, I wish the revised manuscript is now sufficiently improved to deserve publication in the Special Issue of IJMS on "Semaphorins, Plexins, and Neuropilin Functions in the Tumor Microenvironment".

Yours sincerely,

Cristiana Angelucci

Round  2

Reviewer 1 Report

The author made extensive modification to their manuscript and tried to respond to all the problems or suggestion improving its quality. I recommend the publication of the work submited.

Author Response

Dear Editor and Reviewers,

Please find enclosed our revised manuscript entitled “Multifaceted Functional Role of Semaphorins in Glioblastoma” [ijms-478374] by C. Angelucci, G. Lama, and G. Sica, for publication consideration in the Special Issue "Semaphorins, Plexins, and Neuropilin Functions in the Tumor Microenvironment" of the International Journal of Molecular Sciences.

We highly appreciate your very careful review of the paper and your detailed valuable comments and suggestions that ensued.

On behalf of all the co-authors, I wish the revised manuscript is now sufficiently improved to deserve publication in the Special Issue of IJMS on "Semaphorins, Plexins, and Neuropilin Functions in the Tumor Microenvironment".

Yours sincerely,

Cristiana Angelucci

Reviewer 2 Report

The manuscript dealing with semaphorins in glioblastoma has been extensively revised. There are only some minor concerns:

The abbreviation “TAM” placed just below or even in the Fig. 1 may be not clear to the reader not very familiar with the issues concerning tumor macrophages. The full term “tumor-associated macrophages” appears for the 1st time on the page 8 (l. 39), but this abbreviation appears also in the Fig. 2 and Tab. 1 (page 4). Therefore, it is recommended to use the full term “tumor-associated macrophages” in the legend to the Fig. 1. It will probably impose the use of the full term of “CSC” (although the full term is used on the 1st page, l. 39), and “NK” (the meaning of this abbreviation, used for the 1st time in the Fig. 2 (page 4) and then on the page 5 (Tab. 1), is probably much more well-known; however, the full term “natural killer” appears for the first time on the page 9, l. 17).

There is still the trouble with the font size – it seems that different sizes were used throughout the manuscript. Although most likely it is beyond the scope of the authors to fix it, please examine carefully the paper proofs, otherwise this bigger fonts will focus the readers’ attention on the inappropriate phrases.

Author Response

Dear Editor and Reviewers,

Please find enclosed our revised manuscript entitled “Multifaceted Functional Role of Semaphorins in Glioblastoma” [ijms-478374] by C. Angelucci, G. Lama, and G. Sica, for publication consideration in the Special Issue "Semaphorins, Plexins, and Neuropilin Functions in the Tumor Microenvironment" of the International Journal of Molecular Sciences.

We highly appreciate your very careful review of the paper and your detailed valuable comments and suggestions that ensued. A further revision of the manuscript has been carried out according with your recommendations. We hope you find our changes satisfactory. Please find below our responses (normal font) to the Reviewer’s comments (italic font).

Reviewer #2:

The manuscript dealing with semaphorins in glioblastoma has been extensively revised. There are only some minor concerns:

·         The abbreviation “TAM” placed just below or even in the Fig. 1 may be not clear to the reader not very familiar with the issues concerning tumor macrophages. The full term “tumor-associated macrophages” appears for the 1st time on the page 8 (l. 39), but this abbreviation appears also in the Fig. 2 and Tab. 1 (page 4). Therefore, it is recommended to use the full term “tumor-associated macrophages” in the legend to the Fig. 1.

As recommended by the Reviewer, the full term “tumor-associated macrophage” has been included in the legend to the Figure 1.

·         It will probably impose the use of the full term of “CSC” (although the full term is used on the 1st page, l. 39), and “NK” (the meaning of this abbreviation, used for the 1st time in the Fig. 2 (page 4) and then on the page 5 (Tab. 1), is probably much more well-known; however, the full term “natural killer” appears for the first time on the page 9, l. 17).

According to the Reviewer’s requests, the full terms “cancer stem cell” and “natural killer cell” have been added to the legends to the Figures 1 and 2, respectively.

·         There is still the trouble with the font size – it seems that different sizes were used throughout the manuscript. Although most likely it is beyond the scope of the authors to fix it, please examine carefully the paper proofs, otherwise this bigger fonts will focus the readers’ attention on the inappropriate phrases.”

We sincerely apologize for the recurrence of this problem. We continue not to understand the cause of this defect. We carefully checked one more time that a single font (Times New Romans) is used throughout the manuscript.

On behalf of all the co-authors, I wish the revised manuscript is now sufficiently improved to deserve publication in the Special Issue of IJMS on "Semaphorins, Plexins, and Neuropilin Functions in the Tumor Microenvironment".

Yours sincerely,

Cristiana Angelucci